# Nursing Practice and Telehealth in School Health Services: A Scoping Review

**DOI:** 10.3390/healthcare11243124

**Published:** 2023-12-08

**Authors:** Raquel Ayuso Margañon, Maria Llistosella, Sonia Ayuso Margañon, Marta Rojano Navarro, Núria Bou Gràcia, Amalia Sillero Sillero

**Affiliations:** 1Mar Nursing School (ESIMar), Parc de Salut Mar, University Pompeu Fabra, 08003 Barcelona, Spain; rayuso@esimar.edu.es (R.A.M.); marta.rojano@fje.edu (M.R.N.); amalia.sillero@eug.es (A.S.S.); 2Social Determinants and Health Education Research Group (SDHEd), Hospital del Mar Medical Research Institute (IMIM), 08003 Barcelona, Spain; 3Primary Health Care, Consorci Sanitari de Terrassa, 08227 Barcelona, Spain; 4Department of Public Health Nursing, Mental Health and Perinatal Nursing, Facultat d’Infermeria, Universitat de Barcelona, 08907 L’Hospitalet de Llobregat, Spain; soniaayusom@ub.edu; 5Primary Health Care Florida Nord, Institut Català de la Salut, 08905 L’Hospitalet de Llobregat, Spain; 6Department of School Nursing, Jesuits Education Foundation, 08010 Barcelona, Spain; 7Department of School Nursing, Sant Gervasi Jesuit Education Foundation, 08006 Barcelona, Spain; nuria.bou@fje.edu; 8Department of Nursing, Escoles Universitàries Gimbernat, 08174 Barcelona, Spain

**Keywords:** school nursing, telehealth, nursing practice, school health, scoping review

## Abstract

Background: The COVID-19 pandemic has propelled the adoption of telehealth in school settings, emphasising the pivotal role of nurses. This review explores the last decade’s evidence on telehealth interventions in school nursing practice; Methods: Following Joanna Briggs Institute guidelines, we conducted a systematic search in PubMed, CINHAL, and Web of Science in March 2023. Out of 518 articles across 21 journals, 32 satisfied the review criteria. The selection process rigorously adhered to PRISMA-ScR guidelines for scoping reviews; Results: The results were categorised into three main areas: (a) the purpose of telehealth and intervention strategies, (b) the role of nursing in school-based telehealth practice, and (c) perceived benefits and limitations of school-based telehealth studies. Telehealth interventions encompass health promotion, mental health management, and early diagnosis. School nurses play a multifaceted role, including management, education, and remote monitoring. While telehealth offers advantages like improved health and cost savings, challenges include digital literacy, device access, and costs; Conclusion: This review underscores the crucial role of telehealth in schools for enhancing healthcare delivery in educational settings. However, more empirical evidence is required to specify nurses’ contributions to school-based telehealth interventions. Promoting their leadership through stakeholder collaboration is essential. Further research should address challenges and opportunities in school nursing practice, enriching healthcare in educational settings.

## 1. Introduction

Telehealth, defined as the provision of medical care at a distance [1], is not a new concept; interest in telehealth within the scientific community was growing before the COVID-19 pandemic, and the need to implement this tool in health interventions has been identified [2,3]. Opinions regarding its efficacy vary widely, with concerns raised about establishing a therapeutic connection due to the absence of face-to-face interactions [4], limitations in conducting comprehensive physical examinations [5], and potential discrepancies in access, posing challenges for specific population subsets [6].

However, the undeniable merit of telehealth has been substantiated by its substantial socioeconomic benefits for patients, families, healthcare professionals, and the healthcare system at large. Notably, it enhances the relationship between professionals and patients, particularly in chronic disease management [7,8]. This modality offers greater job security to professionals by facilitating information accessibility and inter-service coordination [9,10]. Advocating for its significance, the National Association of School Nurses (NASN) underscores the value of telehealth in augmenting access to school and community health services, suggesting its potential to streamline health support for school children while minimising interruptions to academic pursuits [11]. Telehealth at school helps schoolchildren focus on learning, facilitates access to different types of care, and reduces time away from classes as well as parental travel [12].

The context of the COVID-19 pandemic was a catalyst for monumental shifts in education delivery worldwide [13]. This circumstance incited paradigm shifts in nursing roles within schools, catalysing the emergence of novel care modalities and transforming how nurses interact with students and families [14,15]. The pandemic necessitated the swift launch of telehealth services, especially critical for monitoring and supporting schoolchildren confined to their homes [2].

During the pandemic, school nurses were compelled to pivot routine services into remote delivery [16], confronting exacerbated chronic issues such as childhood obesity [17], sedentary behaviours [18], and complications in managing diabetic patients [19]. Additionally, they undertook newer responsibilities brought about by the pandemic, including infection control [20], development of evidence-based educational materials [21], and personal care and homeschooling. School nurses became pivotal figures, ensuring a safe learning environment by implementing prevention strategies, symptom management, testing procedures, and contact tracing, extending support not only to students but also to their families and school staff [22].

Various studies highlight the leadership role of school nurses in preventing infection and controlling the pandemic in the school population [23,24]. Their interventions improved communication between the home, school, and healthcare providers [25]. Children and adolescents were one of the groups most affected by the psychosocial impact of the pandemic [26,27,28], which led to mental health complications such as hyperactivity [29], bad behaviour problems, aggression and agitation [30], sleep disorders [31,32], mood disorders, and self-harming behaviours [33]. School nurses were key to guiding families and the rest of the educational community, constituting a physical, emotional, and social safety net [22].

However, integrating telehealth into school settings predates the pandemic, prompting an essential inquiry into how nurses can deliver safe, high-quality care through this medium. While the utility of telehealth in general health services has been extensively explored, studies elucidating the application of telehealth in school health services and the role of school nurses in these practices remain relatively limited. Several experts advocate for comprehensive research in this domain [34,35].

While literature reviews concerning the usefulness of telehealth in generalist services [36] and its application among young individuals outside the school setting [37] exist, there is a notable absence of reviews examining the scope and depth of evidence about telehealth interventions specifically within the school environment.

The overarching questions guiding this scoping review are as follows: (a) What telehealth intervention strategies have been implemented in school health services, particularly concerning the COVID-19 pandemic? (b) What role did nurses have in the practice of telehealth? (c) What benefits and limitations were associated with these interventions? 

The primary objective of this exploratory review is to ascertain the scope and nature of evidence concerning telehealth interventions in school nursing practices over the last decade and to explore the extent to which nurses can deliver safe and high-quality care through telehealth.

## 2. Materials and Methods

### 2.1. Study Design

The scoping review method provided us with the opportunity to incorporate diverse methodologies (i.e., quantitative and qualitative studies) to identify the nature and extent of our research [38]. This scoping review adhered to the guidelines specified in the Joanna Briggs Institute Reviewer’s Manual [39] and followed the PRISMA-ScR (Preferred Reporting Items for Systematic Reviews and Meta-Analyses Extension for Scoping Reviews) checklist to ensure methodological rigor [40]. The completed checklist is provided as Appendix A.

### 2.2. Eligibility Criteria

Eligibility criteria were estimated a priori. The eligibility criteria were predetermined and included a comprehensive range of research designs, encompassing original research articles, both qualitative and quantitative, without geographical limitations. Eligible study designs comprised randomised controlled trials, nonrandomised controlled trials, pre- and post-test studies, various observational studies, case reports, and qualitative investigations. This review considered documents published in English and Spanish that delineated telehealth strategies and interventions in school health services. The age range considered for participants was from 6 to 18 years. Additionally, this review did not exclude studies involving children whose ages fell outside the specified range but were relevant to interventions in school health services. Nor were documents referring to children enrolled in special education schools within these age ranges excluded.

Exclusion criteria were defined to exclude articles in languages other than English or Spanish, documents not primarily focused on telehealth in school settings, and those falling outside the realm of empirical research. Systematic or scoping reviews, short communications, letters to the editor, and brief articles were intentionally excluded from our scoping review. Additionally, exclusion criteria were applied to articles with no access to the full text, which may have arisen due to limited availability or restricted access. These criteria were implemented to uphold the quality and depth of our review.

### 2.3. Search Strategy

The search strategy was conducted in articles published over the last 10 years, from March 2013 to March 2023, across three prominent databases: PubMed, CINAHL, and Web of Science. PubMed and Web of Science are two of the most popular bibliographic databases for life sciences and biomedical disciplines and cover most of the medical literature [41]. CINHAL is one of the most comprehensive English language databases indexing the best nursing literature, including publications from the American Nurses’ Association and the National League for Nursing and the most widely used source of nursing research worldwide. Another reason for choosing these databases is that they offer a user-friendly interface with basic and advanced search functions [42]. Keywords were meticulously translated into MeSH terms, ensuring comprehensive search coverage by validating synonyms and “equivalence relations”. In instances where specific MeSH terms were not available, relevant keywords were incorporated to ensure total search outcomes. The search was limited to the past ten years to review the most current interventions.

To ensure inclusivity and a holistic approach to the study topic, grey literature from esteemed school nursing organisations and websites was included in the search, encompassing sources such as NASN, Colombian School Nursing Network, Chilean Nursing Society and School Health (SOCHIESE), Syndicate National de Infirmières Conseillères de Santé (SNICS), ACEESE/ACISE, AMECE, Australian Nursing and Midwifery Federation (Victorian Branch), and Scientific Society of School Nursing of Cantabria (SOCEEC).

The search strategy was a refined combination of the three primary concepts: (1) telehealth, (2) school health service, (3) school nursing. This strategy was corroborated by two senior information specialists (M.C.L. and A.S.S.) and was fine-tuned to focus on identifying telehealth interventions feasible for nurses within school settings.

The search strategy used included the following MeSH terms and keywords (MEDLINE/PUBMED): (“Telenursing”[MeSH Terms] OR “Telemedicine”[MeSH Terms]) AND (“Schools”[MeSH Terms] OR “School Nursing”[MeSH Terms] OR “School Health Services”[MeSH Terms]).

The detailed search strategy encompassing MeSH terms and keywords for all four databases is delineated in Appendix A.

### 2.4. Quality Assessment

We refrained from conducting a critical appraisal or bias assessment of the articles included in our scoping review [43]. Article selection was carried out by independent researchers who adhered to predetermined eligibility criteria. The selected studies conformed to the levels of evidence for effectiveness defined by the Joanna Briggs Institute. The level of evidence available was high for most articles. Seventeen (53.12%) studies were identified as level 1 evidence, three (9.37%) studies were identified as level 2 evidence, three (9.37%) studies were identified as level 3 evidence, eight (25%) studies were identified as level 4, and one article (3.12%) was not identified as any level, as there is no mention of the qualitative method in JBI Levels of Evidence [44]. See Table 1.

### 2.5. Data Extraction

Following the search, all identified citations were systematically organised and uploaded to the bibliographic management software MendeleyCite©2022. Subsequently, these records were imported into the online systematic review tool Rayyan©2022. In this phase, researcher author R.A.M. led the process by meticulously excluding duplicate documents and articles with incorrect publication dates or those not meeting the specified study design criteria.

Following this initial curation, the titles and abstracts of the remaining articles were reviewed by researchers R.A.M. and S.A.M. to assess their alignment with the predefined inclusion criteria. Concurrently, researchers M.L.P. and A.S.S. verified the excluded studies to ensure the appropriateness of the exclusion criteria applied.

For the articles that passed the initial screening, the full text was carefully assessed against the inclusion criteria by researchers N.B.G., S.A.M., and M.R.N., with responsibilities shared equally. To ensure data integrity, approximately 20% of each author’s extractions were randomly cross-checked.

Any disagreements that arose during the various stages of the selection process were resolved through consultation with an additional reviewer. The search results and the process of study inclusion were reported following the PRISMA-ScR guidelines for updated scoping reviews [40].

To extract the studies included in the review, a Microsoft Excel table (Microsoft Corp., Redmond, WA, USA) was employed. This enabled the systematic tabulation and categorisation of the gathered information based on the review’s objectives, encompassing details such as publication title, author, country, year, study type, the age group of the studied population, the purpose of telehealth utilisation, the role of nursing, as well as the benefits and limitations identified in the development of the intervention, including any interventions related to the COVID-19 pandemic. The data extraction tool was continuously reviewed and adjusted to enhance accuracy and consistency.

### 2.6. Analysis and Synthesis of Data

The analysis was conducted using Microsoft Excel and SPSS software (version 26.0; IBM Corp., Armonk, NY, USA). We employed descriptive statistics, focusing on frequencies and proportions, to synthesise the gathered data.

## 3. Results

### 3.1. Search Outcomes

The search strategy resulted in 518 publications; after excluding duplicates, which were screened by title according to the eligibility criteria, 391 articles remained. They were further filtered by abstract, resulting in 95 articles. After the full-text review of these remaining articles, sixteen were rejected due to failed retrieval and one relevant article from the grey literature was included. Thirty-two reports remained for the final review. Reasons for exclusion of full text were as follows: design (n = 8); no telehealth (n = 5); wrong context (n = 8); out of age range (n = 13), parent/adult study (n = 12); duplicate article (n = 2). The PRISMA flow diagram of the study selection process is shown in Figure 1.

### 3.2. Characteristics of Studies

Among the 32 articles incorporated into this review, the studies exhibited a diverse array of characteristics. Notably, fifteen (46.87%) studies employed a randomised controlled trial (RCT) design, two studies employed a prospective experimental design, while three (9.37%) adopted a quasi-experimental design, including one pre–post design (3.12%) and two (6.25%) studies using a prospective evaluation methodology. An additional three (9.37%) studies were categorised as observational analytical studies. A total of eight (25%) studies were classified as descriptive observational studies, including cross-sectional study (1), case series (2), individual case report (1), and not specified (4). Lastly, one (3.12%) study employed a qualitative methodology.

Regarding the age of participants, twelve (37.50%) studies were centred on children aged 6 to 12 years, while nine (28.12%) studies focused on adolescents aged 13 to 18 years. Notably, 11 (34.37%) studies encompassed interventions targeting both age ranges.

The geographical distribution of these studies showcased a diverse landscape, with eleven (34.37%) studies originating from the United States, five (15.62%) from Australia, four (12.5%) from the United Kingdom, and six (18.75%) conducted in Canada, Sweden, and the Netherlands. Additionally, the countries in which single studies were carried out were Bangladesh, New Zealand, Greece, Honk Kong, Poland, and South Africa. Most articles were published in high-income countries according to the WHO regional classification (83.33%).

To address the research questions effectively, the results were thoughtfully categorised into three categories: (a) the purpose of telehealth and intervention strategies, (b) the role of nursing in school-based telehealth practice, and (c) perceived benefits and limitations of school-based telehealth studies. Furthermore, it is noteworthy that nurses played a pivotal role in the interventions in eight (25%) of the studies, with two studies contextualised explicitly within the framework of the COVID-19 pandemic. These studies provided valuable insights into how telehealth was adapted and leveraged to address the unique needs of students and their families during this extraordinary period, as detailed in Table 1.

### 3.3. Purpose of Telehealth and Intervention Strategies

Within the 32 studies included, 14 different intervention focuses were identified. These interventions areas encompassed various health and educational objectives, as follows:(a)Hearing screenings and speech and communication problems in five studies (15.62%) [48,60,66,72,74];(b)Promoting healthy habits and lifestyles, with a focus on healthy eating and physical activity in four studies (12.5%) [49,53,55,61];(c)Promoting mental health and emotion management in four studies (12.5%) [45,52,70,73];(d)Prevention and management of respiratory problems, especially asthma, in four studies (12.5%) [50,59,67,69];(e)Oral health screening and hygiene in three studies (9.37%) [51,56,65];(f)Prevention of tobacco, alcohol, and other substance use in two studies (6.25%) [57,76];(g)Reproductive health and pregnancy prevention in two studies (6.25%) [54,63];(h)Screening and monitoring of mental health problems in two studies (6.25%) [46,75];(i)Care and behaviour in one study (3.12%) [68];(j)Prevention of psychosocial problems in one study (3.12%) [56];(k)Autism assessment in one study (3.12%) [47];(l)Diagnosis and management of skin scabies in one study (3.12%) [71];(m)Prevention of skin cancer and photoaging in one study (3.12%) [64];(n)Screening for metabolic syndrome in one study (3.12%) [58].

Intervention strategies across these studies comprised tele-education using customised information content in capsule format in fifteen studies (46.87%) [46,49,51,52,53,54,57,61,62,63,64,69,73,75,76]; telediagnosis or telemonitoring using wearable devices to detect signs, improve treatment adherence, and prevent complications in eight studies (25%) [48,55,56,58,65,66,72,74]; and teleconsultation (telehealth) in nine studies (28.12%) [45,47,50,59,60,67,68,70,71]. Notably, all these interventions were implemented within the context of school health services or school settings.

No specific interventions directly related to the COVID-19 pandemic were identified, although one study claimed that the pandemic prompted the telehealth intervention [46], and another study was conducted during the pandemic [45].

Additionally, no interventions were specifically designed for special education (refer to Table 2 for a concise summary of these findings).

### 3.4. Role of Nursing in School-Based Telehealth Practice

In this review, nursing played a vital role in 25% of the telehealth interventions, while in the rest of the articles, the role of nurses was not reported, as it was mostly a paediatric medical team who developed the interventions. The nursing interventions highlighted in this context encompassed the following:-Administration and monitoring of medical guidelines via teleconsultation to ensure efficient completion; support to school staff; and scheduling and monitoring attendance of telemedicine visits in the prevention and diagnosis of asthma [50,71]. In addition, the nurses helped ensure seamless communication with the physician during virtual visits; served as guardians of students’ health information; and operated as the primary interface between parents, students, paediatricians, and teachers on school health issues [67];-Motivational interviews with adolescents through teleconsultation focusing on specific risk areas and mental health in particular [75]. Behavioural assessment analysed in opportunistic screening for psychosocial problems through YouthCHAT [56];-Finally, nurses collaborated by conducting educational interventions via telemedicine and conducting comprehensive asthma education aimed at preventing crises, as well as assessing treatment efficacy and monitoring crises via teleconsultation [69]. In the detection and follow-up of hearing problems, they coordinated with primary care staff in the implementation of the Tele-Health-Kids (THK) programme and collaborated in follow-ups via telemedicine [74]. They coordinated and executed the biochemical analyses (blood draw) and anthropometric tests for metabolic syndrome screening via telemedicine [58].

For detailed information on these nursing interventions, please refer to Table 2.

### 3.5. Perceived Benefits and Limitations of School-Based Telehealth Studies

The integration of digital tools into educational settings has demonstrated many benefits in improving students’ heath. The studies included in this review highlighted various advantages, such as the following:

Complementing traditional interventions: Digital tools were found to complement traditional health interventions, leading to improved health outcomes and better control of problems such as anxiety [45,52], oral pathology [62], asthma [50], speech and language problems [60], and hearing problems [72]. They also proved effective as a screening method for hearing problems [66], resulting in fewer symptoms, better health problem management, and reduced readmissions and visits to the respiratory emergency department [67];

Enhanced clinical care and family cooperation: Some authors reported superior clinical care and increased cooperation with families [61,74]. Children actively engaged in their health [55,62,76], promoting a positive public health impact [71]. In addition, remote care enabled improved monitoring of health problems in rural communities with difficult access to health services and immigrant populations with language difficulties [46,47,65,70];

Knowledge and skill acquisition: Digital tools facilitated the acquisition of knowledge, skills, and positive health attitudes [49,54] They were also instrumental in reducing or preventing risk behaviours, including unintended pregnancy [63], photoaging [64], and smoking cessation [57]. In some instances, they resulted in cost savings within the US healthcare system, known for its free market care model [59], and in healthcare system in general [56,68]. Additionally, they served as an information system for nurses regarding the health status of vulnerable school children, enhancing face-to-face consultations [75];

Unclear or insignificant outcomes: However, it is worth noting that in four of the included studies, the benefits of the outcomes were unclear or showed no significant change [53,58,69,73].

Despite these substantial benefits, there were several limitations observed in the practice of school-based telehealth:

Study design limitations: Some limitations were inherent to the study designs, which might have led to contamination between participant groups, information bias, carry-over effect, or challenges in instrument validation [51,60,66];

Limited generalisation: Many studies had limitations related to sample size and short trial durations, making it challenging to generalise their findings beyond the studied context [45,46,48,54,55,56,57,58,59,61,63,64,67,68,74,75];

Digital literacy and access: Limited digital literacy among students, families, teachers, and healthcare workers was identified as a challenge [51], along with restricted access to mobile phones in some territories [76]. Technical difficulties and limited resources were a barrier in various studies [49,50,55,70]. Implementation costs and the need for government confidence in the strategy were also recognised [71];

Other limitations: Finally, other limitations included lack of participation in the programme [52,53,69,72] and concerns regarding data confidentiality [63].

Notably, two reviewed articles reported no limitations [47,65].

These findings demonstrate the potential of school-based telehealth, underscoring the importance of addressing associated challenges and limitations.

## 4. Discussion

The scoping review of 32 studies conducted to examine the current breadth and range of research on nurses’ use of telehealth interventions in school settings over the past decade has revealed several crucial insights.

First, these interventions have as their primary objective the promotion of healthy lifestyle habits and enable the early identification and management of various health problems in different age groups [77]. Within this realm, mental health has become a critical concern that can be effectively addressed in the school setting using telehealth. This focus aligns with the World Health Organisation’s recent prioritisation of emotional well-being and health in children and adolescents, particularly pertinent given that one in seven children and adolescents suffers from a mental disorder. This concern has been further compounded by the challenges imposed by the COVID-19 pandemic [78,79].

A critical area highlighted in our review concerns the management of chronic diseases such as asthma and addressing speciality care issues, such as hearing loss, through telehealth. Studies have shown that regular communication facilitated by telehealth tools such as video conferencing, mobile telephones, and patient portals significantly helps manage chronic childhood diseases such as asthma [80,81,82]. Noteworthy is the role of school nurses, who are critical in coordinating healthcare and education through telehealth [83]. Their participation ensures effective follow-up by primary and speciality care providers, reducing fragmented care and unnecessary medical services [84,85].

Despite the apparent value of telehealth during the COVID-19 pandemic, there remains a dearth of research on telehealth in schools [86,87]. The challenge is to conduct primary studies with comprehensive data, especially when schools remain closed.

Regarding the strategies used, various telehealth methods have been reported, including teleconsultation through videoconferencing, telediagnosis, telemonitoring through portable devices, and tele-education through digital tools or mHealth for health promotion and the prevention of risk behaviours. Although specific information on the most effective telehealth strategies in schools remains limited, some studies suggest that mHealth appeals to children and adolescents due to its ease of access to health information and services [88,89,90]. Other authors highlight the effectiveness of paediatric teleconsultation, enabling flexibility in terms of space and scheduling, time savings, and prevention of self-diagnosis and self-medication [91,92].

Overall, there is consensus among researchers that telehealth holds significant potential to improve health outcomes for children and adolescents [93]. Moreover, the National Association of School Nurses (NASN) recognises telehealth as an alternative delivery model. It underscores its potential to enhance the impact of school nurses on student health and academic outcomes [94].

It is worth noting that our review identified limited research regarding the utilisation of artificial intelligence (AI) in the school environment and telehealth practice. However, some researchers have highlighted the advantages of AI in this domain, such as identifying health patterns, providing valuable health information, and enabling professionals to make more informed and accurate decisions [95,96]. The future applications of AI in school-based telehealth practice promise further advancements.

In summary, our findings emphasise the significant role of nurses in telehealth, offering essential remote healthcare and enhancing school students’ access to healthcare services. Nurses play a collaborative and proactive role, working closely with physicians and other healthcare professionals. However, it is noteworthy that, according to our review, nurses tend to prefer preventive and health promotion activities over providing care during acute exacerbations. This preference aligns with other authors who underscore the fundamental role of nurses in simplifying access to and utilisation of telemedicine services [97].

Furthermore, nurses facilitate education and follow-up with other healthcare services for schoolchildren with special health needs, particularly during the COVID-19 pandemic, particularly in underserved or geographically challenging communities within the school context [98,99]. It is essential to acknowledge that nurses have also led telehealth interventions beyond the school setting [100,101,102].

In our exploration of school-based telehealth, we encountered limited evidence about the role of nursing within school settings. This necessitated a reevaluation of our search methodology, prompting a strategic shift towards interventions led or actively participated in by nurses. Moreover, within the array of included studies, the benefits of the outcomes were occasionally indistinct or exhibited no significant changes in specific instances [53,58,69,73]. Some studies revealed inherent limitations tied to study designs, information bias, or challenges in instrument validation [51,60,66]. Furthermore, several studies grappled with constraints related to sample size and the brief duration of trials, posing challenges in generalising their findings [45,46,48,54,55,56,57,58,59,61,63,64,67,68,74,75].

In summary, while our exploration sheds light on the potential of school-based telehealth, it also underscores critical evidence gaps and methodological limitations that warrant careful consideration in shaping future research endeavours in this evolving field.

### 4.1. Limitations and Strengths

This scoping review has several limitations which may have influenced the results obtained. We only included articles written in English or Spanish and with full text available, which could have led to the loss of some critical articles in this review. In addition, we performed the bibliographic search in three databases, and although these databases are highly recommended according to JBI methodology for scoping reviews, relevant sources of information may, however, have been omitted, and the review question depends on what information is available. Thus, the study’s main limitation was challenges in delineating the role of nursing in the context of telehealth within school settings. While we recognise their involvement, limited evidence led us to reevaluate our search methodology, focusing on interventions led by nurses or those in which they actively participated.

Categorising articles by intended age ranges posed another limitation. Initially, we planned to divide articles into two age groups: 6–12 years and 13–18 years. However, due to the variability in age ranges covered in the articles, we chose to conduct separate analyses for studies encompassing both age groups, ultimately considering the age range of 6–18 years to examine the available information and optimise overall results comprehensively.

Despite the limitations mentioned above, this study presents strengths to be taken into account. Our study is based on a comprehensive scoping review that includes a wide range of sources and a substantial number of studies, which provides a thorough overview of the subject matter.

We followed established guidelines and checklists, such as the Joanna Brigs Institute Reviewer’s Manual and the PRSMA-ScR checklist, to ensure the quality and transparency of our review.

The selection and data extraction stages were conducted with the active participation of at least two independent reviewers, enhancing the reliability and rigor of the study.

In summary, while our study offers a comprehensive overview of the subject, these strengths and limitations should be taken into consideration when interpreting our findings and framing future research in this area.

### 4.2. Implications of Findings

Indeed, there are implications for practice, research, and policy based on the study’s findings, which are as follows.

#### 4.2.1. Implications for Practice

Enhanced access to healthcare: This study highlights the potential for improved access to healthcare through virtual consultations in school settings. Students can readily receive advice, diagnoses, and health monitoring, which is especially beneficial in remote geographic areas where traditional healthcare access may be limited;

Early detection of health problems: Telehealth offers a valuable tool for detecting various health issues, including visual, hearing, developmental, or emotional disorders. This early identification can lead to timely interventions and support for students;

Health promotion and disease management: Nurses play a crucial role in telehealth by providing information on healthy habits, disease prevention, and the management of chronic conditions. This empowers students to adopt a healthier lifestyle and take control of their well-being.

#### 4.2.2. Implications for Research

Exploring artificial intelligence (AI) applications: This study opens the door for future research into integrating AI into healthcare within educational settings. Investigating how AI can enhance the delivery of healthcare and support students’ health and well-being is a promising avenue for further inquiry.

#### 4.2.3. Implications for Policy

Integrating telehealth into school healthcare: Policymakers should take note of the potential of telehealth in school settings. This study can catalyse and encourage the adoption of measures that incorporate telehealth into educational institutions, ensuring that students receive comprehensive care and support. Active leadership of nurses: Policymakers should also consider measures that encourage operational leadership by nurses in telehealth initiatives within schools. Collaborative efforts involving all relevant stakeholders, including educational institutions, healthcare providers, and government bodies, can ensure the effective implementation of telehealth.

## 5. Conclusions

In conclusion, this scoping review delves into a decade of telehealth practices in school settings, spotlighting the integral role of nursing professionals. Of 32 studies, 15 adopted robust randomised controlled trial designs, focusing on themes like auditory screening, mental health promotion, and respiratory management.

The age-centric approach encompassed children (6–12 years) and adolescents (13–18 years), reflecting a global distribution of studies. The results, categorised into telehealth purpose, nursing role, and perceived outcomes, highlighted nurses’ critical involvement in 25% of interventions across diverse health domains.

The perceived benefits derived from these interventions were substantial, augmenting traditional approaches, enhancing clinical care, and fostering collaborative efforts within families.

The implications underscored the enhancement of healthcare access, early anomaly detection, and the elevation of nursing leadership within the telehealth domain.

These findings seamlessly align with the review’s objectives, underscoring the pivotal role of telehealth in schools. While progress has been made, ongoing exploration and a reinforced emphasis on nursing leadership remain imperative in this dynamic landscape.

In this context, the challenges and opportunities observed in school nursing practice present promising avenues for future research—focusing on leadership and innovation to cultivate healthier, digitally inclusive educational environments.

## Figures and Tables

**Figure 1 healthcare-11-03124-f001:**
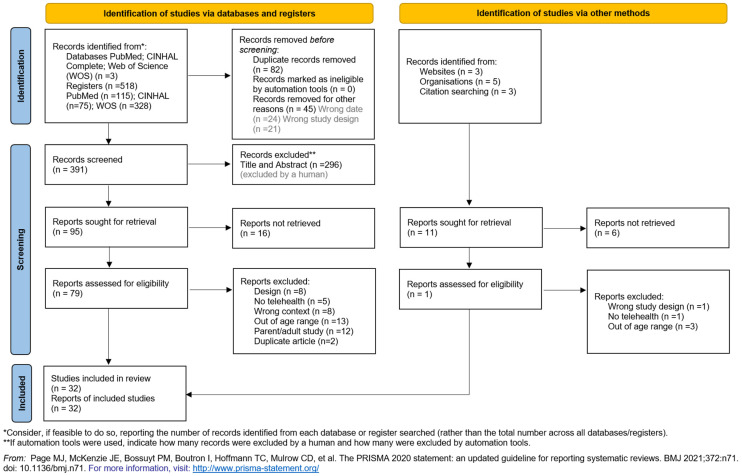
PRISMA 2020 flow diagram for new systematic reviews which included searches of databases, registers, and other sources, adapted from [40].

**Table 1 healthcare-11-03124-t001:** Main results of the included studies.

Reference	Design and JBI Level of Evidence *	Data Sample	Purpose of Telehealth Use	Nursing Role	Benefits Perceived	Limitations Perceived
Green et al. (2023) [45]	Descriptive observational (uncontrolled case series). Level 4.c—Case series	UK. Children aged 8–9	To provide a brief, therapist-guided treatment called Online Support and Intervention (OSI) to parents/carers of children identified through school-based screening as likely to have anxiety problems. Strategy: teleconsultation	N/A	Session-to-session improvements in all measures. The online platform was found to be more convenient and less stigmatising than traditional face-to-face clinic appointments	Difficulty of extrapolation and limitations of study design
Martinez et al. (2023) [46]	Qualitative method. N/A—No mention of qualitative method in JBI Levels of Evidence	USA. Immigrant youth aged 13–18	To examine the impact that the transition to telehealth had on a school-based group prevention programme for immigrant youth, FUERTE Program. Strategy: tele-education through teleconference	N/A	Improved access to mental healthcare for immigrant youth using telehealth	Lack of representativeness of the population and difficulty in extrapolating the data
Shahidulla et al. (2022) [47]	Quasi-experimental. Level 2.c—Quasi-experimental prospectively controlled study	USA. School-aged youth (13–18) in under-resourced school systems in central Texas	To increase resources for autism evaluation and support for under-resourced schools through a state-wide school telehealth initiative. Strategy: teleconsultation	N/A	Increases access to timely autism spectrum disorder (ASD) assessments and support through a statewide school telehealth initiative	Not reported
Emmett et al. (2022) [48]	Randomised controlled trial. Level 1.c—RCT	Alaska (USA) aged 4–21. All children enrolled in the Bering Strait School District	Improve specialised follow-up time after positive school hearing tests (in possible hearing loss or ear disease). Strategy: telemonitoring through mHealth	None of the schools have school nurses	Improving early access to specialists for rural children, reducing health disparities	Short duration of the trial and small sample size due to age restrictions in school achievement tests and hearing-related quality of life questionnaires
Brown et al. (2021) [49]	Experimental design. Prospective impact assessment. Level 1.d—Pseudo-RCTs	Australia. Children aged 6–12	To assess the impact of different lunchbox messages on parents’ intention to prepare a healthy lunchbox. Strategy: tele-education through mHealth—short message service (SMS)	N/A	Particularly strong effects on behavioural intentions in relation to messages related to other Health Belief Model (HBM) constructs	Technical limitations in the use of technology
Crabtree-Ide et al. (2021) [50]	Quasi-experimental. Level 2.c—Quasi-experimental prospectively controlled study	USA. Children aged 3–10	To assess whether a telemedicine-based programme for enhanced asthma management (SB-TEAM), designed to overcome barriers to care for families of urban school-aged children, can be financially sustainable in urban school settings. Strategy: teleconsultation	Nurses oversaw the telehealth programme, assistants, and telemedicine visits programme. School nurses identified children with uncontrolled asthma at the beginning of the school year, administered daily preventive asthma medication, and monitored symptoms	The use of telemedicine to improve asthma control in underserved communities was shown to be highly effective	Economic limitations and lack of resources
Marshman et al. (2021) [51]	Observational action-research. Level 3.e—Observational study without a control group	UK. Students aged 11–16	To describe the development process of a behaviour change intervention to improve the oral health of students. Strategy: tele-education through mHealth (SMS)	N/A	The intervention was needed to address the shortage of oral health promotion interventions for secondary school students and attempted to integrate a traditional classroom delivery method complemented with a newer mHealth technology solution. Parents and teachers were involved in the process	Restriction on the number of characters available; the need for basic literacy; and limited access to mobile phones for some young people
Moltrecht et al. (2021) [52]	Observational exploratory trial. Level 4—Not specified	UK. Children aged 10–12	To explore the use of a new app-based intervention designed to support children’s emotion regulation in schools. Strategy: tele-education through mHealth—mobile application (app)	N/A	Using app improves emotion regulation; making app available improves access to teachers	Teachers report lack of content as one of the main obstacles to implementing the application
Sutherland et al. (2021) [53]	Cluster randomised controlled trial. Level 1.c—RCT	Australia. Children aged 7–12	To assess the effectiveness of a multicomponent, mobile health-based intervention (SWAP IT) in reducing the energy contribution of discretionary foods and drinks packed for children to consume at school. Strategy: tele-education through mHealth (SMS/app)	N/A	There were no significant changes in pupils’ engagement in school	Technical problems were detected that could have reduced participation
Ahmed (2020) [54]	Quasi-experimental. Level 2.d—Pretest–post-test or historic/retrospective control group study	Bangladesh. Adolescent girls aged 14–19	To assess the effect of an mHealth tool on knowledge regarding reproductive health. Strategy: tele-education through mHealth (SMS)	N/A	Knowledge score increased to 70.8% ± 9.7% after the 8-week SMS intervention on reproductive health knowledge	Difficult to extrapolate
Langlet et al. (2020) [55]	Descriptive observational acceptability–usability study. Level 3.e—Observational study without a control group	Sweden. Adolescents aged 13–18	To conduct a formative evaluation of a smartphone app for monitoring daily meal distribution and food selection. Strategy: telediagnosis/telemonitoring through mHealth (app)	N/A	The smartphone application has a high acceptability and usability among students	Lack of resources (shortage of devices). Difficulty in extrapolating
Thabrew et al. (2020) [56]	Cluster randomised controlled trial. Level 1.c—RCT	New Zealand. Adolescents aged 14–15	To assess the effectiveness of repeated psychosocial screening of high school students using the help assessment tool Please check the accuracy. Strategy: telediagnosis/telemonitoring through mHealth (app)	Conducting the YouthCHAT assessment and managing positive detection of analysed behaviour	Its availability in schools, with school nurses and counsellors, facilitates its application for opportunistic and routine psychosocial assessment, reducing costs compared to time-consuming face-to-face assessments	Difficult to generalise data. Lack of data on the satisfaction of participating nurses
Müssener et al. (2020) [57]	Randomised controlled trial. Level 1.c—RCT	Sweden. High school students aged 17	To estimate the effectiveness of a novel mHealth intervention called NEXit Junior, which targets smoking cessation. Strategy: tele-education through mHealth (SMS)	N/A	The results of the NEXit junior trial demonstrated a positive effect on smoking cessation of a text messaging-only intervention	Follow-up was assessed by self-reported measures and was not biochemically verified
Bacopoulou et al. (2019) [58]	Descriptive observational study. Level 4—Not specified	Greece. Adolescents aged 12–17	Detect MS (International Diabetes Federation criteria) and explore its associations with anthropometric, sociodemographic, and behavioural parameters using telemedicine in the school setting. Strategy: telediagnosis/telemonitoring	Coordinate and execute the biochemical analysis of the study (blood extraction), as well as the anthropometric tests	N/A	Difficulty in extrapolating data in different contexts
Bian et al. (2019) [59]	Randomised controlled trial. Level 1.c—RCT	USA. Children aged 3–17 enrolled in the Medicaid Statistical Information System	To examine the associations of a school-based telehealth programme with all-cause emergency department (ED) visits. Strategy: teleconsultation	N/A	21% reduction in the likelihood of ED visits among a subsample of children with asthma; programme was associated with an overall reduction in ED visits of more than 20%	Limited generalisation and difficult extrapolation
Langbecker et al. (2019) [60]	Experimental prospective evaluation study. Level 1.d—Pseudo-RCTs	Australia. Children aged 3–12 in rural schools	To assess a service delivering speech and language therapy (SLT) and occupational therapy (OT) via videoconferencing. Strategy: teleconsultation	N/A	The majority of children who received telehealth services (SLT and OT) through the Health-e-Regions programme showed steady improvement over time	Limitations of the design (lack of validation strategies)
Lau et al. (2019) [61]	Cluster randomised controlled trial. Level 1.c—RCT	Hong Kong. Adolescents aged 12–16	To assess the effectiveness of a short message service (SMS) intervention on promoting physical activity. Strategy: tele-education through mHealth (SMS)	N/A	El efecto recordatorio mediante SMS a los participantes es la función más obvia para lograr efectos significativos de la intervención y proporciona apoyo motivacional, interactivo y social	Difficulty to generalise due to small sample size
Marshman et al. (2019) [62]	Randomised controlled trial. Level 1.c—RCT	UK. Young people aged 11–13 from deprived areas	The Bringing Information and Guided Help Together (BRIGHT) trial assessed the clinical use and cost-effectiveness of a behaviour change intervention—classroom-based session (CBS) embedded in the curriculum and a series of SMS delivered to participants twice daily to remind them to brush their teeth—compared to usual curriculum and no SMS. Strategy: tele-education through mHealth (SMS)	N/A	Motivational SMS improved self-reported oral health	Contamination between participating groups
Tebb et al. (2019) [63]	Cluster randomised control trial. Level 1.c—RCT	USA. Latin adolescents aged 13–18	To assess the implementation of a mobile health contraception decision support intervention in school-based health centres. Strategy: Tele-education through mHealth (app)	N/A	Preventing pregnancy in at-risk communities, facilitating access to contraceptive information through the Health-E You mobile digital tool	Difficulty in generalising to other environments. Limitations related to technological infrastructure (reliable connectivity, confidentiality assurance, communication required across multiple levels)
Brinker et al. (2018) [64]	Randomised controlled trial. Level 1.c—RCT	Brazil. Students in secondary schools aged 13–18	To assess the effectiveness of a photoaging intervention for skin cancer prevention delivered by medical students in secondary schools. Strategy: tele-education through mHealth (app)	N/A	Raises awareness among future doctors about the importance of skin cancer prevention and influences the improvement of health behaviours in the prevention of photoaging	Difficulty in extrapolating data in different contexts
Estai et al. (2018) [65]	Cross-sectional study. Level 4.b—Cross-sectional study	Australia. Children and adolescents aged 5–14 from low-risk areas	To develop a resource reallocation model for school dental screening that takes advantage of teledentistry. Strategy: telediagnosis/telemonitoring	N/A	Telemedicine improves access to dental healthcare	Not reported
Govender et al. (2018) [66]	Randomised controlled trial. Level 1.c—RCT	South Africa. Children aged 6–12	To assess the efficacy of an asynchronous telehealth-based service delivery model using automated technology for screening and diagnostic testing. Strategy: telediagnosis/telemonitoring	N/A	Automated asynchronous telehealth-based automated hearing tests in the school context can be used to facilitate early identification of hearing loss	Limitations related to “carry-over effects” where a participant’s performance in one test may influence his or her performance in the other test
Halterman et al. (2018) [67]	Randomised controlled trial. Level 1.c—RCT	USA. Children aged 3–10	To assess the effect of the School-Based Telemedicine Enhanced Asthma Management (SB-TEAM) Program. Strategy: teleconsultation	Review telemedicine visits to ensure efficient completion of guideline-based care, including appropriate prescribing of preventive medications. Nurses do not receive additional compensation	Children in the SB-TEAM group had more symptom-free days every 2 weeks post intervention compared to children in the enhanced usual care group and were less likely to have an emergency department visit or hospitalisation for asthma	Difficulties in ensuring double-blinding and difficult extrapolation
McLennan (2018) [68]	Descriptive observational study. Level 4—Not specified	Canada. Children aged 6–12	To assess the effectiveness of video-conferencing telehealth linkage attempts to schools to facilitate mental health consultations. Strategy: teleconsultation	N/A	Mental health consultations were successfully conducted through two different telehealth video conferencing links between a health centre and several schools in this pilot initiative. The linkage that was able to utilise existing hardware in both the health and school systems holds promise for scalability given the low equipment costs and minimal technical support required	Difficulty of extrapolation
Perry et al. (2018) [69]	Cluster randomised controlled trial. Level 1.c—RCT	USA. Children (aged 7–14) living in an impoverished rural region	To examine the effect of a school-based asthma education programme delivered by telemedicine. Strategy: tele-education through videoconference	Nursing is a component of the study, receiving telemedicine educational intervention, comprehensive asthma education, and interactive question and answer sessions aimed at preventing asthma attacks, assessing treatment efficacy, and managing attacks	Although some behavioural changes were observed among intervention participants, these were insufficient	Reluctance of primary care providers to change or initiate asthma medication according to recommendations without a formal referral to a clinical setting; possible lack of contact between families and primary care providers due to difficulty accessing care within the 3-month period in which the primary outcome was measured; low caregiver participation in educational sessions; low response rates to the survey
Pradhan et al. (2018) [70]	Descriptive observational study. Level 4—Not specified	USA. Vulnerable youth aged 12–17	To describe a telehealth model for delivering integrated mental health services in a telemedicine-based school health clinic. Strategy: teleconsultation	N/A	Access to specialised care for the most vulnerable young people increased	Connected teams that maintain communication and facilitate follow-up to interventions and mitigate cultural and distance barriers
Zettler-Greeley et al. (2018) [71] (grey literature)	Descriptive observational study (individual case report). Level 4.d—Case study	USA. Students aged 6–12	To describe how telehealth stopped a contagious outbreak at a school. Strategy: teleconsultation through telenursing	School nurses are positioned to triage complaints appropriately, reducing the potential for overuse, a common concern given the easy access to care offered by telehealth. The nurse ensures seamless communication with the physician during virtual visits; serves as a gatekeeper of students’ personal health information; and operates as the primary interface between parents, students, paediatricians, and teachers on school health issues	Telehealth visits are more cost-effective than an emergency room visit and can help prevent and contain public health problems. Telehealth in schools reduces absenteeism, increases instructional time, and offers benefits to underserved children in regions where access to healthcare is limited	Unresolved political issues related to licensing, jurisdiction, and reimbursement hinder the growth of paediatric telehealth
Skarzynski et al. (2016) [72]	Analytical observational study. Level 3.e—Observational study without a control group	Poland. Children aged 7–8	To validate hearing screening procedures in young children and collect data using a telemedicine model. Strategy: telediagnosis/telemonitoring	N/A	Improved monitoring and control of hearing problems through telemedicine	Limitation in the collection of data through questionnaires
Burckhard et al. (2015) [73]	Randomised controlled trial. Level 1.c—RCT	Australia. Students aged 12–18	To examine the feasibility of an online school-based positive psychology programme (Bite-Back) delivered in a structured format over a 6-week period utilising a workbook to guide students through website content and interactive exercises. Strategy: tele-education through interactive activities	N/A	No significant results were reported	Problems in the implementation of Bite Back in the school setting.
Langkamp et al. (2015) [74]	Descriptive study of a case series. Level 4.c—Case series	USA. Children and adolescents aged 3–21	Detection of hearing problems through the Tele-Health-Kids (THK) telemedicine tool. Strategy: telediagnosis/telemonitoring	Performing nursing duties and coordinating with primary healthcare staff for the implementation of the THK programme	High level of parental satisfaction with the programme; increased adherence to treatment and cooperation from children; families felt more actively involved in their children’s healthcare; the familiar figure of the school nurse instils confidence and security in the family and children; reduced stress for the child and parents; increased likelihood of successful medical examination; superior and more effective clinical care than traditional office visits	Sample size too limited to make generalisations about telemedicine use
Bannink et al. (2014) [75]	Cluster randomised controlled trial. Level 1.c—RCT	Netherlands. Students aged 13–18	To assess the effect of E-health4Uth and consultation on well-being (mental health status and health-related quality of life) and health behaviours (alcohol and drug use, smoking, safer sex). Strategy: tele-education through mHealth (SMS)	School nurses were trained to conduct motivational interviews with adolescents aged 15–16 years. During the consultation, they focused on specific risk areas and mental health. Adolescents were referred to another professional if deemed necessary	E-health4Uth allows the selection of vulnerable adolescents and provides nurses with information about the health of these adolescents. It contributes to the efficiency of face-to-face consultations	Abandonment and difficulty of extrapolation
Cremers et al. (2014) [76]	Cluster randomised controlled trial. Level 1.c—RCT	Netherlands. Children aged 10–11	To assess whether email and mobile phone prompts stimulate primary school children to reuse an internet-delivered smoking prevention intervention. Strategy: tele-education through mHealth (SMS)	N/A	Prompts can encourage children to reuse an intervention website aimed at smoking prevention	Limited access to technology and tedious measures to ensure confidentiality that complicate participation

Note 1: N/A = Not applicable; RCT = randomised controlled trial. Note 2: * Level of evidence for effectiveness, developed by JBI Levels of Evidence and Grades of Recommendation Working Party, October 2013. https://jbi.global/sites/default/files/2019-05/JBI-Levels-of-evidence_2014_0.pdf (accessed on 29 October 2023).

**Table 2 healthcare-11-03124-t002:** Purpose of telehealth, intervention strategies, and role of nursing in school-based telehealth practices.

Purpose of Telehealth	N (%)
Hearing screenings and speech and communication problems	5 (15.62%)
Promoting healthy habits and lifestyles (healthy eating and physical activity)	4 (12.50%)
Promoting mental health and emotion management	4 (12.50%)
Prevention and management of respiratory problems such as asthma	4 (12.50%)
Oral health screening and hygiene	3 (9.37%)
Prevention of tobacco, alcohol, and other substance use	2 (6.25%)
Reproductive health and pregnancy prevention	2 (6.25%)
Screening and monitoring of mental health problems	2 (6.25%)
Prevention of psychosocial problems	1 (3.12%)
Autism assessment	1 (3.12%)
Diagnosis and management of skin scabies	1 (3.12%)
Prevention of skin cancer and photoaging	1 (3.12%)
Screening for metabolic syndrome	1 (3.12%)
**Intervention strategies**	
Tele-education	15 (46.87%)
Teleconsultation (telehealth)	9 (28.12%)
Telediagnosis or telemonitoring	8 (25%)
**Nursing role**	
Administration and monitoring of medical guidelines via teleconsultation; scheduling and monitoring of telemedicine visits; facilitating fluid communication in virtual visits; and mediating between different agents (family, teachers, paediatricians, and students).	3 (9.37%)
Motivational interviews through teleconsultation on specific risk areas and mental health; behavioural assessment in the detection of psychosocial problems via YouthCHAT.	2 (6.25%)
Comprehensive asthma education aimed at crisis prevention; evaluation and treatment of asthma crises; coordination with the primary care team for the application of the hearing problems detection programme; follow-up through telemedicine; and coordination and execution of anthropometric tests for the detection of metabolic syndrome.	3 (9.37%)

N (%): number of articles reported (percentage). Percentage calculated on the total of articles included in the review (N = 32).

## Data Availability

The datasets generated and/or analysed during the current study are not publicly available but are available from the corresponding author on reasonable request.

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
