# Peer review of "Nursing Practice and Telehealth in School Health Services: A Scoping Review"

_healthcare, 2023, doi:10.3390/healthcare11243124_

Round 1

Reviewer 1 Report

Comments and Suggestions for Authors

Dear authors,

 I am pleased to have reviewed your paper. Despite numerous studies on this topic, I must emphasize my satisfaction with your research. It is an exceptional review that provides crucial information for future researchers. The introductory section provides valuable insights into what follows. The well-written methodology and clear PRISMA enhance the significance of the study. While Table 1 is extensive, it offers valuable information. The discussion, study limitations, and conclusion are well-executed and written. The searched databases provide valuable information.The main drawback lies in its extensive length. I suggest condensing tables and shortening sections to reduce the overall page count.

Best regards.

Author Response

Point 1: Introduction. I suggest condensing tables and shortening sections to reduce the overall page count.

Response 1: Thank you very much for your comments and suggestions, we are grateful for the possibility of improving  this section. We have reorganised the tables to shorten the document (page 16, lines 313-318).

Reviewer 2 Report

Comments and Suggestions for Authors

The study design is appropriate

What was the reason for not using other databases? Why did researchers use only these three databases?

The reason for choosing the research methodology should be clearly defined.

In what time frame were the articles selected?

Short communication, letter to the editor and brief article, or lack of access to the full text of the articles, how was it managed?

Please discuss the why of the results in the discussion section.

What is the limitation of the present study?

Author Response

Point 1: Methods. What was the reason for not using other databases? Why did researchers use only these three databases?

Response 1: We appreciate your suggestion. We have reviewed this section and have added the reason for the database selection. We added:

‘PubMed and Web of Science are two of the most popular bibliographic databases for life sciences and biomedical disciplines and cover most of the medical literature [44]. CINHAL is one of the most comprehensive English language databases indexing the best nursing literature, including publications from the American Nurses' Association and the National League for Nursing, and the most widely used source of nursing research worldwide. Another reason for choosing these databases is that they offer a user-friendly interface with basic and advanced search functions [45]’. (lines 136-144).

Point 2: Methods. The reason for choosing the research methodology should be clearly defined.

Response 2: Thank you for your suggestion. The scoping review method allowed us to incorporated diverse methodologies (i.e. quantitative and qualitatives studies) to identify nature and extent of research. Clarification has been made in the manuscript on page 3, lines 108-110.

Point 3: Methods. In what time frame were the articles selected?

Response 3: Thank you for your appreciation, we have added the time frame in which the articles were selected. We added: ‘The search strategy was conducted over the last 10 years, from March 2013 to March 2023, through three prominent databases: PubMed, CINAHL and Web of Science.' (lines 135-138).

Point 4: Methods. Short communication, letter to the editor and brief article, or lack of access to the full text of the articles, how was it managed?

Response 4: Thank you very much for your suggestion, we have added it on page 3, lines 130-134.

Regarding the exclusion criteria, short communication, letters to the editor, and brief articles were intentionally excluded from our scoping review. This decision was made to maintain the focus on comprehensive and substantive primary studies, ensuring a thorough exploration of the subject matter. Additionally, exclusion criteria were applied to articles needing more access to the full text, which may arise due to limited availability or restricted access. These criteria were implemented to uphold the quality and depth of our review.

As for managing these criteria, our team meticulously assessed each potential article against the predefined criteria during the screening and selection process. Articles falling under the categories of short communication, letters to the editor, brief articles, or full-text access were excluded from the final analysis.

Studies without access to the full text were identified as "Reports not retrieved". The short communication, letter to the editor and short article were excluded in the "Title and abstract" section. See Figure 1. PRISMA 2020 flow diagram for new systematic reviews, which included searches of databases, registers and other sources. (lines 216-218).

Point 5: Discussion. Please discuss the why of the results in the discussion section.

Response 5: We appreciate the valuable feedback from the reviewer, and we have considered the suggestion to elaborate further on the "why" of the results in the discussion section.

In our exploration of school-based telehealth, the limited evidence regarding the role of nursing within school settings emerged as a focal point. This scarcity of evidence prompted us to reevaluate our search methodology, leading to a strategic pivot towards interventions specifically led or actively participated in by nurses.

The discussion delves into the challenges faced in uncovering substantial evidence, touching upon the occasional indistinctness or lack of significant changes in outcomes observed in certain instances [53, 58, 69, 73]. Importantly, we acknowledge the intrinsic limitations inherent in the study designs, information bias, and challenges associated with instrument validation [51, 60, 66]. Moreover, we highlight the constraints linked to sample size and the brief trial durations in several studies, underscoring the difficulties in generalizing findings across diverse contexts [45-46, 48, 54-59, 61, 63-64, 67-68, 74, 75].

Our discussion illuminates the potential of school-based telehealth and underscores the critical evidence gaps and methodological limitations. By explicitly addressing these challenges, we aim to provide a comprehensive understanding of the intricacies of researching and implementing telehealth interventions in school settings. This acknowledgement informs the foundation for future research endeavours, emphasizing the need for a nuanced and context-specific approach in advancing this evolving field.

Clarification has been made in the manuscript on page 18, lines 421-432.

Point 6: Limitations. What is the limitation of the present study?

Response 6: Thank you for your question, there is a section 4.1 on pages 19, lines 433-443: Limitations and strenghts. We added: ‘This scoping review has several limitations, which may influence the results obtained. We only included articles written in English or Spanish and with full text available, which could lead to the loss of some critical articles in this review. In addition, we performed the bibliographic search in three databases, and although these databases are highly recommended. However, according to JBI methodology in the Scoping reviews, relevant sources of information may be omitted, and the review depends on information on the available review question. Thus, the study's main limitation was challenges in delineating the role of nursing in the context of telehealth within school settings. While we recognize their involvement, limited evidence, led us to reevaluate our search methodology, focusing on interventions led by nurses or those in which they actively participated’.

We have reviewed the wording to improve comprehension of the limitations and strenghts section (lines 433-449).

Reviewer 3 Report

Comments and Suggestions for Authors

Author Response

Point 1: Methods. It lacks a PICO question, which would guide the study.

Response 1: Thanks for your comments on formulating the scoping review question. We appreciate the importance of providing a clear framework to guide the study, and we would like to offer additional insight into our approach. In scope reviews, you can follow this method to ask extracted from

https://jbi-global-wiki.refined.site/space/MANUAL/4687737/11.2.2+Developing+the+title+and+question

In developing the primary question for our scoping review, we aimed to ensure it aligned with the title and incorporated the essential Population, Concept, and Context (PCC) elements. The primary question: 

  1. a) "What telehealth intervention strategies have been implemented in school health services, particularly concerning the COVID-19 pandemic?"

This question addresses the PCC elements comprehensively while aligning with our scoping review's objectives. This overarching question provides a robust foundation for our study, guiding the protocol's development and enhancing the efficiency of our literature search.

While our primary question may stand alone in specific scoping reviews, we acknowledge the potential benefits of including sub-questions. Sub-questions can offer a nuanced exploration of particular attributes related to Context, Population, or Concept, contributing to a more detailed evidence mapping.

We sub-questions exploring the role of nurses in the practice of telehealth or the benefits and limitations associated with these interventions have been considered:

(b)"What role did the nurse have in telehealth?"

"(c) What benefits and limitations were associated with these interventions?"

These sub-questions provide depth and precision to our scoping review, enhancing the exploration of telehealth intervention strategies in school health services during the COVID-19 pandemic.

We hope this clarification addresses your concerns, and we appreciate the opportunity to refine our approach based on your insightful feedback.

The questions detailed above were not in question format in the manuscript. They have been clarified by putting them in question format. See page 3, lines 98-102.

Point 2: Methods. Assuming what the authors expressed, that they followed the guidelines specified in the Joanna Briggs Reviewer's manual, it is mandatory to fit the selected studies with the levels of evidence defined by the Joanna Briggs Institute. And this must be evident in the tables presenting the studies.

Response 2: Thank you very much for the suggestion. We certainly believe that his comment improves the quality of the manuscript. We have clarified this aspect in the Quality Assessment section. We have added:

‘The selected studies conformed to the levels of evidence for effectiveness defined by the Joanna Briggs Institute. The Level of evidence available was high for most articles. 17 (53.12%) studies were identified with level 1 evidence, 3 (9.37%) studies were identified with level 2 evidence, 3 (9.37%) studies were identified with level 3 evidence and 8 (25%) studies were identified with level 4, and 1 article (3.12%) was not identified with any level, as there is no mention of the qualitative method in JBI Levels of Evidence [45]’. (lines 168-174). We have also evidenced this in Table 1. We added the ‘Level or evidence JBI’ element as a heading in the design. (See Table 1, lines 250-256).

Reviewer 4 Report

Comments and Suggestions for Authors

I found this to be a very interesting paper. However, please consider the following points.

In the abstract, (1), (2), (3), and (4) are unnecessary and should be revised accordingly.

How about rearranging Table 1 in the order of the research methods described in "3.2. Characteristics of studies"?

The results were categorized into three main areas: [a] telehealth purposes and strategies, [b] the diverse role of school nurses in telehealth, [c] recognized benefits and limitations.

Regarding [b] the diverse role of school nurses in telehealth, Table 1 shows that 22 of the 32 articles do not describe the role of nursing. Therefore, it can be assumed that the results of this study are not available.

[c] recognized benefits and limitations.

From the Table 1, it should be "[c] perceived benefits and limitations of the study".

The paper is 25 pages long.

Is it necessary to include Tables 2 through 4?

Authors should be able to do concise writing without the use of tables.

The authors have reviewed many papers, but I do not believe that the results are adequately integrated at this time.

I hope this helps.

Author Response

Point 1: Abstract. In the abstract, (1), (2), (3), and (4) are unnecessary and should be revised accordingly.

Response 1: Thank you for your suggestion, it has been removed. See Abstract section

Point 2: Methods. How about rearranging Table 1 in the order of the research methods described in "3.2. Characteristics of studies"?

Response 2: Thank you very much for the suggestion. The criteria we used to order Table 1 were the year of publication of the article from most current to oldest and authorship in alphabetical order as the second criterion. We are very grateful for your suggestion and consider this a good strategy for ordering the table. According to your suggestion, in the 'Design' column we have added the Joanna Briggs Institute's levels of evidence for effectiveness. This column is now called ‘Design and Level of Evidence JBI'. Also, we have reordered the columns 'Design and Level of evidence JBI' and 'Data sample' to make the table easier to read. (See Table 1, lines 250-256).

In addition, we adapted the wording of the ‘3.2. Characterístics of studies’ section makes the review easier to read and more transparent (lines 219-228).     

Point 3: Results. The results were categorized into three main areas: [a] telehealth purposes and strategies, [b] the diverse role of school nurses in telehealth, [c] recognized benefits and limitations.

Regarding [b] the diverse role of school nurses in telehealth, Table 1 shows that 22 of the 32 articles do not describe the role of nursing. Therefore, the results of this study are not available.

Response 3: Thank you very much for the suggestion. In the section ‘Role of nursing in school based telehealth practice’, we added: ‘In this review, nursing played a vital role in 25% of the telehealth interventions, while in the rest of the articles, the nursing role was not reported, being mostly a pediatric medical team who developed the interventions’. (Lines 292-295).

Furthermore, we have alluded to this issue in the ‘4.1. The Limitations and Strenghts section, adding that it was the study's main limitation, for which we had to reevaluate our methodological strategy. (Line 440-443).

Point 4: Results. [c] recognized benefits and limitations. From the Table 1, it should be "[c] perceived benefits and limitations of the study".

Response 4: Thank you for your suggestion; it has been modified in the manuscript and the abstract for better consistsency. (Table 1, Lines 250-256).

Point 5: Results. The paper is 25 pages long. Is it necessary to include Tables 2 through 4?

Authors should be able to do concise writing without the use of tables.

Response 5: Thank you for your suggestion, for a better understanding we have proceeded modified table 2 by integrating table 3 and deleting table 4. See Table 2, lines 317-318

Point 6: Conclusions. The authors have reviewed many papers, but I do not believe that the results are adequately integrated at this time.

Response 6:  In response to this valuable input, we enhanced the integration of results to provide a more cohesive and unified conclusion. We have structured the results in a single Table (see Table 2). We have removed the conclusion, and we have added the new conclusion in lines 492-514. 

Round 2

Reviewer 2 Report

Comments and Suggestions for Authors

The answers and reasons given by the authors were appropriate and acceptable

Author Response

Point 1: The answers and reasons given by the authors were appropriate and acceptable.

Response 1: We appreciate the reviewer's comments and are glad to have resolved the suggestions for improvement.

Reviewer 4 Report

Comments and Suggestions for Authors

I think this manuscript has been properly corrected.

Please consider the following two points.

1) There are some mistakes (spaces, abbreviations, fonts) in Table 1, so please correct them.

2) Regarding the conclusion, I think that lines 490-501 are appropriate and clear conclusions for the purpose of this research.

However, in line 502, it seems to reopen the discussion.

The sentences in lines 502-511 seem to have lowered the quality of this paper.

How about deleting these sentences, lines 502-511?

I hope this helps.

Thank you very much.

Author Response

Point 1: Table 1. There are some mistakes (spaces, abbreviations, fonts) in Table 1, so please correct them.

Response 1: Thank you for your suggestion. We appreciate the reviewer's comment and we are grateful for the possibility of improving this section. We reviewed the Table 1 again:

- we reviewed the punctuation marks.

- we reviewed formatting (matched , typeface and font size, line spacing, and eliminated white spaces).

- we reviewed the abbreviations and added the meaning.

Point 2: Conclusions. Regarding the conclusion, I think that lines 490-501 are appropriate and clear conclusions for the purpose of this research.

However, in line 502, it seems to reopen the discussion.

The sentences in lines 502-511 seem to have lowered the quality of this paper

Response 2: We are very grateful for the Conclusions proposal. We removed the lines proposed by the reviewer and adapted the wording of this section acording to this suggestion. We recognize that it facilitates reading and the new proposal gives strength to the article. (lines 501-509).
